# Integrated Remote Sensing and 3D GIS Methodology to Strengthen Public Participation and Identify Cultural Resources

**Dimitris Goussios [1] and Ioannis Faraslis [2],***

1 Department of Planning and Regional Development, University of Thessaly, Pedion Areos, 38334 Volos, Greece
2 Department of Environmental Sciences, Campus Gaiopolis, University of Thessaly, 41110 Larissa, Greece
* Correspondence: faraslis@uth.gr; Tel.: +30-2410-684344

**Abstract:** In the context of territorial development, the construction of specific and competitive local resources is based on the identification of their intangible and material elements but also their links to the region. The connection between these links and local heritage, along with their spatial dimension, makes the active participation of residents in the entire process necessary. This paper presents the application of an integrated methodology that fosters the involvement of residents in a process of collecting relevant implicit information, with the assistance of experts, in order to identify cultural resources from different historical periods. This methodology is based on the synergy of three components: interdisciplinarity, local community participation, and the use of non-destructive cutting-edge technologies (remote sensing, UAV mapping, ground-penetrating radar, and 3D GIS interactive representations). The use of various methods and tools is organized in successive phases, the objective being the substantial participation of residents through 3D interactive visualisations of their area. 3D representations enable the activation of local memory in conjunction with the collection of information regarding location, type, and traces of cultural resources. The entire process validates the implicit information that guides the competent authorities and experts in the further search for more precise information, both from satellite data (high-resolution images) and images from subsurface mapping (ground-penetrating radar). The proposed methodology significantly accelerates the process of identifying cultural resources and provides a comprehensive picture to local government and cultural institutions about the area's cultural resources and planning possibilities while reducing the failures and costs of the research process.

**Keywords:** 3D GIS representations; public participation; cultural resources; remote sensing; archaeological traces

## 1. Introduction

Cultural heritage, for decades considered a protection tool and a key element in building a nation [1,2], tends to acquire a developmental function and a resource status in the context of territorial development [3–5]. Furthermore, the Faro Convention (Convention-cadre du Conseil de l'Europe sur la valeur du patrimoine culturel pour la société. Faro, 27.X.2005), emphasises the relationship between cultural heritage and communities and reiterates that, visible or not, it acts as a supporting element to their identity and the specificity of their resources. Activated during the process of creating territorial areas [6,7], heritage establishes that cultural and historical resources can create new forms of local development [8]. Indeed, development initiatives of this scale seem to place cultural heritage at the centre of their territorial plan and actions [9]. Such projects organise the meeting of the economy and heritage through the participation of local actors on the basis of a functional relationship between heritage and territorial resources. Managing this relationship is defined by a form of governance [10].

This dynamic resurgence of cultural heritage raises the issue of its management through its developmental function. Already, new standards (territorial development, social

economy) and new intervention mechanisms (development companies, NGOs) question the traditional ways of managing heritage resources [10]. Furthermore, following the trend of transition from public to common goods, several local collectives participate in the management of such resources as a result of the inability of public management to respond to a collective interest which tends to be defined by the dynamics of territorial development.

The tendency to expand the function of cultural heritage also concerns archaeological resources. Kaeser [11] notes that by providing a material base for collective identification, archaeology can give meaning to space in a territorial area. This means that the archaeological resource is territorial when it concerns a place of memory where a local community finds elements necessary to its identity and the timeless links to the place. These ties are strengthened not so much by the archaeological object itself but by its contribution to production over time and the enhancement of the historical depth of local space (landscape, production, etc.). In other words, archaeological finds managed by archaeologists are not of interest here, as much as the various functions that these may reveal and that the local community may incorporate into new uses. This multifunctionality of heritage if it is connected to local memories and experiences of society within a place of memory, contributes to the distinction between history as experienced locally, and that told by historians. Finally, residents of each area, by attempting a "bottom-up" interpretation of a resource, intervene in the dichotomy of memory/history, rendering the place experienced in the past; a topical phenomenon. This updating is fostered by historical continuity and, therefore, the need to fill any gaps in the cultural resources corresponding to intermediate historical periods. It is, however, noted that the relevant public authorities operate separately in their national and regional planning and are difficult to coordinate in terms of the complete identification of the cultural resources in each territorial area, at the community or municipal level.

Two central issues therefore arise. The first is the issue of temporality and the obstacles to the implementation of local plans to integrate heritage into the process of local resource valorisation, as this depends on the reconciliation of different temporalities: (a) the time-consuming process of enhancing cultural resources based on planning by public authorities and (b) the operational planning of the territorial area. The second issue stems from the need of each region to identify and integrate the elements of its cultural heritage into the process of territorial development. The long-term results of this objective depend on the region's ability to connect its space, the goods it produces, and the services it offers, to the dual, intangible and material, nature of this heritage. However, in the short term, the activation of this heritage as a territorial resource or as an intangible element of these resources requires the active participation of residents, local heritage bodies, and relevant intangible (know-how, knowledge, practices) and material (monuments, landscapes, etc.) information. Residents seem to be the most suitable to integrate elements of cultural heritage into territorial resources as only they can move both ways between the material (visible elements in space such as ruins, spatial arrangements, landscapes) and the intangible nature of heritage, drawing from history and determining its value. It in fact relates to the organisation of a patrimonialisation process which includes: (a) the anchoring of heritage in the place through elements related to the cultural object's contribution to production over time (landscape, production, etc.) and the experiences of the local community, and (b) links between the territorial resource and heritage.

The main issue that arises concerns the way and means of supporting the involvement of local communities in this process, considering that modernisation has transformed the landscape, disconnected the population from the place which functioned as an environmental memory and removed points of reference recognition. There is therefore a need for the process of heritage identification and territorialisation to be organised around real space (what has been preserved) and virtual space (what can be represented). Such a hybrid environment can be the result of a combination of methods and techniques based on new technologies and, as a methodological chain, can help co-produce information in order to identify resources and highlight their links to the region. This combination involves the use of interactive 3D virtual worlds with the interaction of specialists/local actors. The virtual

world can create representations of space and its elements, both in their current and past situation (paths, landscapes, etc.), which function as recognisable reference points for the local community [12].

Such a methodology, integrated into a plan and supporting territorial diagnosis in terms of the connection between heritage and resources, can create the appropriate environment to organise a process of cultural heritage territorialisation. Enhancing the territoriality of cultural heritage through the participation of local actors requires specialised methods and visualisation tools that contribute to the identification of the links between heritage and the region, producing an integrated valorisation of the cultural resources in each territorial area [13,14].

*Modern Research Methods for Cultural Resources*

The new approach to the heritage/territorial resource relationship, by focusing on the transition from the object to its relationship to a space at the centre of human activity, is based on three methodological components:

- Interdisciplinary practice as a means of shaping and applying methodologies in the field. The interface between disciplines such as geography, archaeology, and computer science enables the combination of methodological tools so that cultural traces as well as their links to the territorial area can be identified.
- Supporting the active participation of residents in the production of information regarding their territorial resources, with the contribution of interactive representations and experts [15]. The extensive use of geovisualisation (geographic visualisation), is based on the fact that information on resources has a spatial dimension. However, in terms of archaeological resources, whereas their location enhances their links to the area, it does not in itself determine the spatial dimension of their issues.
- The utilisation of cutting-edge technologies and techniques. Progress in the fields of Geographical Information Systems (GIS), remote sensing, and more specialised technologies such as subsurface mapping applications with ground-penetrating radars (GPRs), was a decisive factor in the development of non-destructive methodologies to identify and valorise cultural resources. More specifically, advanced methods of spatial analysis, image processing (or even from UAVs—unmanned aerial vehicles) and interactive 3D geovisualisations, are increasingly applied in the spatial identification of resources [16–19].

In modern research methods for cultural resources (with a focus on archaeological resources), the combination of remote sensing tools (satellite images/aerial photographs), geophysical methods, topographic studies, and archaeological knowledge, is increasingly integrated with knowledge of farming systems etc. [20,21]. There is particular reference to geophysical surveys because, in contrast to field research, they map the subsoil without destroying the object in question [22]. In general, geoinformatics provides tools which, through non-destructive methods, help detect traces of human interventions in space, minimising costs in time and money. In addition, the use of GIS allows the recording and analysing of geographical data from different sources and at various scales. This contributes to the correlation of information both spatially and temporally.

However, the utilisation of these three components requires, due to the abundance of information, an expansion of available sources as well as the participation of local actors and the integration of various old and new methods into a new, combined, adapted, and effective methodology. This methodology should allow the cross-checking and combination of any kind of source. This enables, according to Michèle Brunet, the integration of cultural resources into broader structures, functions, or networks [23]. In the context of this theoretical and methodological approach, this paper aims to show that the combination of methods and tools that utilise 3D geovisualisation as well as non-destructive technologies (GIS, remote sensing), can effectively support the active participation of residents based on a virtual space that is recognisable to them. A methodological chain is created which fosters the integration of other technological tools, helping accelerate the identification of

cultural heritage and its links to the region and integrating the valorisation of the cultural resources in each territorial area [14,24].

## 2. Methodology

### 2.1. Methodological Challenges

Taking into account the new developments presented above, an innovative methodological chain is proposed, with human capital at its centre, and used to record the historical value of human activity as a spatial arrangement, structure, or cultural resource at a specific site. This methodology is also applied to detect subsurface cultural traces with the use of multiple methods combining social research and cutting-edge technologies. This combination enables investigation in areas that in the past formed the spatial basis for the organisation and operation of settlements as well as production and socio-cultural activities. At the same time, it achieves quick results in recording, mapping, and highlighting the relevant information based on two axes:

- Preparation/support of resident participation, with the aim of activating memory. The region's geographical environment is represented in 3D environment in order to allow residents to project onto space any information memory related to cultural heritage. This process enables the recording of places referred to in residents' memories and can lead to a discovery, as with archaeological excavations, to a representation, a virtual configuration, and navigation in space as it was in the reference period determined by the supplier of information (e.g., before land consolidation) [9].
- The precise location on the ground and the geovisualisation of information using geographic information systems (GIS), remote sensing methods, and geophysical surveys.

The following section details the proposed steps for the complete recording and valorisation of cultural resources in an area, based on the three aforementioned pillars: interdisciplinarity, participation, and cutting-edge technologies.

### 2.2. Proposed Methodological Chain

The complex and combined methodology consists of three interconnected steps with a feedback loop (Figure 1):

- Information collection using conventional methods.
- Mobilisation of the local community to collect information with the contribution of 3D GIS interactive visualisations and mapping.
- Application of geoinformatic methods in mapping and identifying cultural resources.

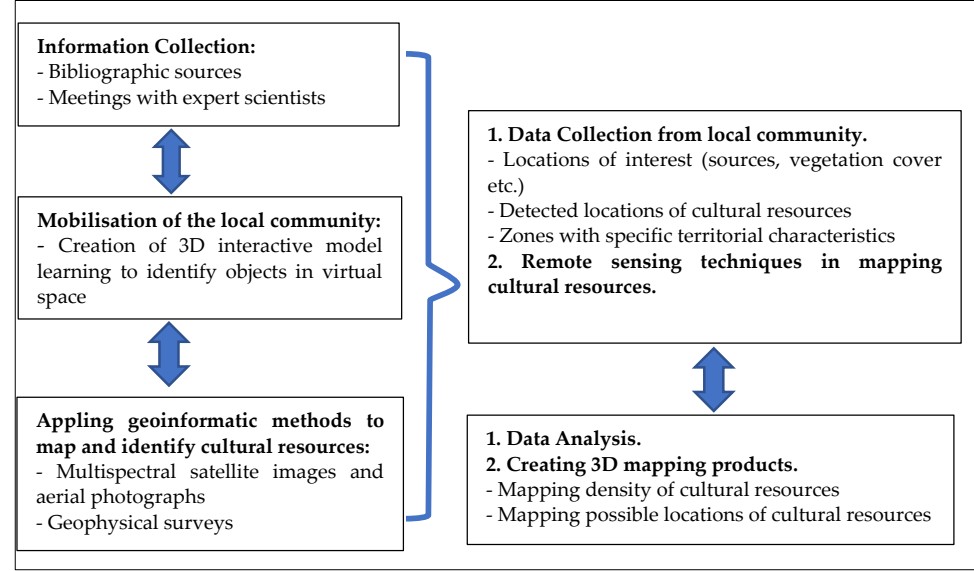

**Figure 1.** Flowchart of the proposed methodology.

　　　Each of the 3 phases of the combined methodology is described below:

### 2.2.1. Information Collection Using Conventional Methods

　　　This phase focuses on the delimitation of the research area, i.e., the identification of the area of interest and the locations of the cultural and archaeological resources and the landscape. For this purpose, 1st level data is collected and mapped. The information is drawn from relevant bibliographic sources as well as knowledge, experiences, and information extracted from experts. Specialist scientists (e.g., archaeologists, historians, etc.), cultural associations, etc. are involved in the process from the start, in order to determine the degree of knowledge regarding the location of the cultural resource. This is followed by:

- Historical research: study of maps, land records, photographs, publications, primary sources, etc.
- Preliminary research: identification-analysis of the various characteristics that compose the landscape of the area's cultural heritage.

### 2.2.2. Mobilisation of the Local Community: Collecting Information with the Contribution of 3D GIS Interactive Visualisations

　　　The second phase involves the mobilisation and participation of the local population in collecting information with the help of 3D modelling. The process includes: (a) consultations with local bodies and (b) development of a reliable and functional 3D model to represent the site.

i.　　Meetings with the local community/residents

　　　This involves a comprehensive "bottom-up" effort to gather information through the mobilisation of local communities considered as bearers of information accumulated through oral tradition and ground-based observations. People who have experienced a space and territorial area are bearers of specific information and witnesses of events, either directly as participants or indirectly as in dialogues on relevant topics with people from older generations. Therefore, they represent an important source of knowledge in identifying resources. In both cases, the research objective is to provoke a two-way path in the interlocutor's timeline through the activation of memory and the representation in space of the information sought. The selection of these interlocutors among members of the local community is preceded by a special survey to identify people with the following characteristics:

- Born at least 10 years before the interventions (land consolidation, change of use, etc.) in the agricultural landscape of the community.
- Observant and with a good sense of direction.
- With a reputation for having knowledge of information related to the local cultural heritage.

　　　Such a trip in space requires a clear description of oral or visual information, accuracy in terms of the points indicated, and confirmation or correction of the information by any other witnesses.

ii.　　Creation of 3D interactive model

　　　The success of the meetings can be enhanced through the application of adapted techniques and tools, such as 3D interactive models (3D GIS), which offer a realistic representation of the landscape. The integration of technical and scientific fields such as 3D interactive geovisualisation, virtual navigation (virtual flight), and remote sensing (using aerial photographs and satellite images) enables the collection, processing, and storage of spatial and descriptive information [25]. The 3D interactive model constitutes a common "language" of communication between researchers and residents, enhancing their dialogue with the following objectives:

- Representation in a way that interlocutors and suppliers of information recognise, with the use of high-definition images, as well as additional elements which act as reference points.
- Reconstruction of the area with the use of past aerial photographs which depict, as accurately as possible, the various past surface material elements (springs, trees, shrubs, etc.) which function as reference points for the older residents of the area.
- To improve the functionality of the 3D interactive model, two main categories of spatial information should be provided:
- Recent cartographic data. Modern high-resolution aerial photographs and satellite images for a first identification of the layout and organisation of the reference area as it is today (roads, infrastructure, relief, etc.).
- Cartographic data from previous years. Use of aerial photographs and bibliographic sources to construct thematic maps from a time before the major material interventions occurred (land consolidation, levelling, clearing, etc.). The aim is to reconstruct the old landscape and restore landmarks in order to facilitate the identification of the information provided by residents.

In essence, the application of 3D representations, by combining different types of information and mapping the changes that have occurred in land cover in recent decades, facilitates navigation in space and time.

### 2.2.3. Application of Geoinformatic Methods to Map and Identify Cultural Resources

#### i. Recording Land Use Cover

In addition to collecting necessary information from residents, the use and processing of appropriate satellite data (as well as aerial photographs) help identify land cover/land use as well as changes that have occurred over time. Essentially, there is a complementarity and synergy between the two different sources. That is, "bottom-up" information and the interpretation of cartographic data and aerial footage (satellite images, aerial photography), achieving a complete and objective representation of elements in space, a better definition of thematic categories, and the spatial determination of relevant densities.

Unlike land cover, land use is not easily identified. To avoid misinterpretation, information on land cover and use should be collected simultaneously. To record changes and classify land cover and land uses, the following codification is used [26,27]:

- Conversion, which refers to the transition from one land cover/land use to another (e.g., from forest to grasslands, from natural vegetation to crops, etc.). More specifically, the information to be collected relates to: (a) vegetation zones, such as forests, scrubland, crops, meadows, etc., (b) the physiography of the area, such as slopes, hydrographic network, wetlands, etc., and (c) man-made structures, such as buildings, infrastructure, etc.
- Dynamic monitoring of changes, which represents the evolutionary course within the land cover/land use category itself (e.g., from dry farming to irrigated crop) due to changes in physical or functional properties. More specifically, the information that will be collected must refer to the intensity of the cultivation systems, the depth of ploughing, the irrigation period, etc.

It is important that the above codifications incorporate the temporal as well as the spatial dimension, the objective being the search for relevant and corresponding sources of information as well as for people who experienced the previous situation and the changes.

#### ii. Multispectral Images and Aerial Photographs as a means of detecting cultural resources

In recent decades, the literature has seen an ever-increasing number of applications that use remote sensing technologies (from multispectral satellite images and aerial photographs) to study, document, and preserve cultural resources [28]. More specifically, in archaeological research, multispectral data have applications in detecting, mapping, and monitoring at different scales. Remote sensing techniques detect indirectly buried archaeological traces. This is achieved through the study of the interaction between matter

and electromagnetic radiation but also knowledge of climatic data and the phenological stages of development of crops located on the surface of the archaeological site [29–32]. Various multispectral data processing methods are applied such as visual interpretation, contrast enhancement, vegetation indices, filtering, etc. [33]. The type and degree of vegetation cover are key elements in image processing. Therefore, the investigation begins with the phenological stages (seasonal changes of vegetation) of the crops grown in the study area [34,35]. The type of cultivation (e.g., cereals), soil and traces which indicate the existence of archaeological findings are analysed and identified in multispectral images [36]. These traces are formed by the visible variations in the density, colour, or height of crops and are separated into "negative traces" or "positive traces" [13,37].

In addition to satellite data, ultra-high resolution images from unmanned aerial vehicles (UAVs) have been used in recent years. The high temporal resolution combined with the ultra-high resolution products they offer are key tools in the detection of elements on the site. Furthermore, UAV surveys can be used for the digital documentation and representation of cultural resources [38,39]. Digitisation offers new tools for recording, consulting, studying, networking, and displaying areas. It paves the way for multiple social, educational, cultural, or economic uses through the implementation and creation of secondary multimedia products. Finally, it facilitates the readability of the areas in terms of inherited and cultural resources.

iii.　Geophysical surveys and the contribution of ground-penetrating radar

The final stage in the proposed methodological chain is the mapping of the subsoil with geophysical surveys and especially with ground-penetrating radar (GPR) technology. It is essentially a non-destructive technology that utilises the different physicochemical properties of subsoil materials to detect anthropogenic structures and more. It has been used for over 40 years in applications in many scientific fields, such as hydrology, geology, edaphology, etc. [40,41]. The literature in particular notes the important role of GPR in Archaeology and in detecting buried archaeological finds [42–44].

The information collected from residents in combination with the analysis of aerial photographs and satellite images, as mentioned in the previous stages of the methodology, leads to the identification of points and zones containing cultural resources, where geophysical surveys are applied, providing direct information regarding the location and size of the cultural trace. The quality of ground-penetrating radar information depends on the condition and building materials (composition) of the underground trace. For the above reasons, mapping is undertaken in close collaboration with expert scientists (Archaeological Service).

Finally, the use of ground-penetrating radar as a complement to the two previous stages (social research, image analysis) completes the implementation of a combined interactive methodology which also constitutes, for all three methods, a non-destructive intervention. This methodological chain allows a gradual and complete process of identifying and valorising the totality of the cultural heritage, in collaboration with the local community, at a relatively low financial cost, further minimising the time needed to valorise resources.

## 3. Case Study Application

### 3.1. Study Area

To apply the proposed methodology, a culturally and archaeologically rich location was chosen: the ancient city of Skotoussa, which is located on the plain of Thessaly (Central Greece) and belongs to the municipality of Farsala (Figure 2). The traces of an ancient city which reached its peak in the 5th century BC are a key visible element of its historical heritage [45,46]. At the same time, the continuous cultivation of high-quality wheat from antiquity until today constitutes a key cultural resource of the region, ensuring its specificity and a potential increase in the added value of its final products. In this context, the primary research objective was to identify these archaeological traces by recording and verifying the testimonies of local residents.

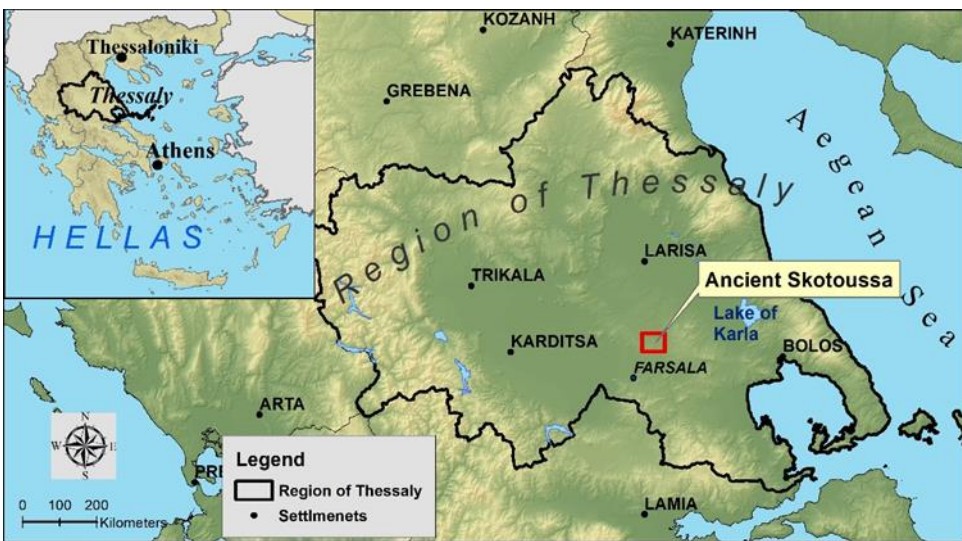

**Figure 2.** Study area, ancient Skotoussa.

The following is a description of the procedure implemented and the results obtained.

### 3.2. Implementation of Methodology and Results

In the first stage of the methodology, information was collected from bibliographic sources, as well as archaeologists about the location of the city, its walls, the location of the acropolis, etc. The second stage involved the collection of information from local residents. To ensure faster and more efficient communication with the local population (mostly farmers), a three-dimensional interactive model (3D GIS) was created in which land cover/land uses were mapped on three different dates: 1945, 1971, and 2018 (Figure 3). The visualisation of the changes that occurred over 75 years was used as a stimulus, especially with older residents, for the purpose of retrieving information from the past. Essentially, the visual stimuli target the activation of voluntary memory.

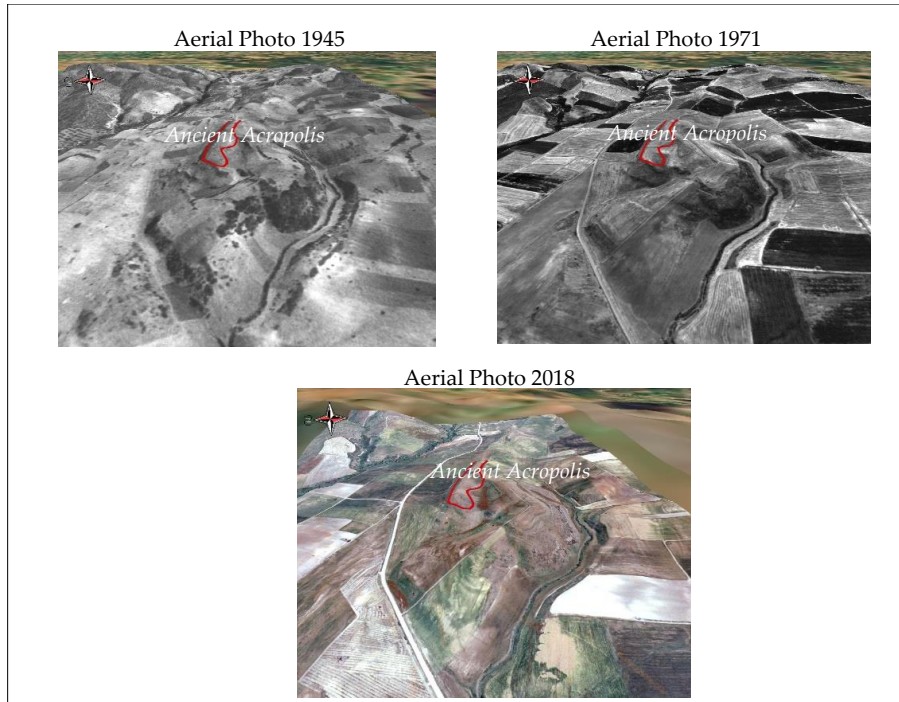

**Figure 3.** 3D geovisualisation of the study area in a 75-year time range.

The changes in land cover/land use on the three different dates are visible. Specifically, the ancient city acropolis is attributed to areas where natural vegetation is located in spacetime (in 1945) followed by conversion into annual crops (in 1971 and 2018). The 1945 aerial photographs depict the condition of land cover before land consolidation. In contrast, the aerial photographs of 1971 and 2018 show the subsequent expansion of farming and the increase in the size of the agricultural parcels. This fact makes it possible to represent the state of the natural and man-made environment and the associated reference points in the middle of the 20th century.

The three-dimensional interactive model of the ancient city was presented, on a high definition screen, to a group of farmers whose farms are located in the area. After a short virtual navigation of the virtual space, they were asked to identify their parcels, as a confirmation of their ability to orientate themselves on the three-dimensional map (Figure 4).

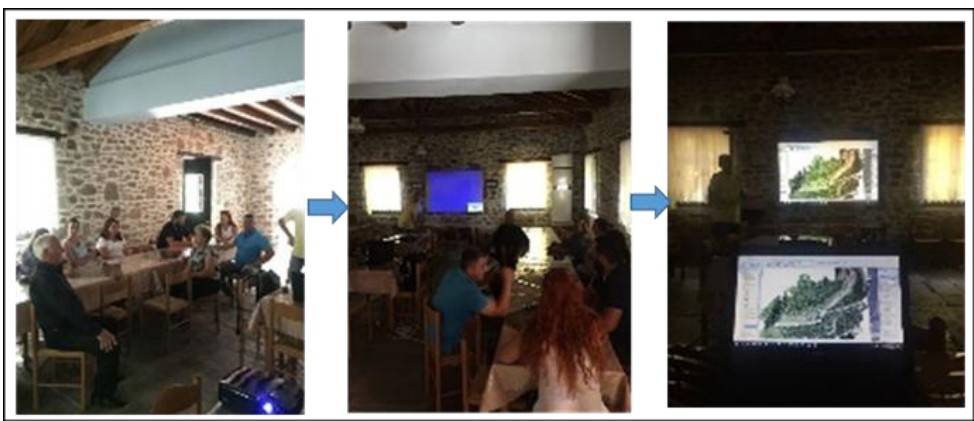

**Figure 4.** Public participation procedures.

The farmers initially were reluctant to report any information. However, after familiarizing themselves with the 3D interactive model, they started pointing out places where ancient traces were found. It is noteworthy that in the case a farmer indicated a position, the rest of them either confirmed or corrected him as to its accuracy. An additional very interesting point was their mutual aid orientation, in the case that an alternation occurred between the 3D renderings of the old (1945) and the contemporary (2018) spatial data.

Next was an attempt to record, in the 3D model, information regarding the existence of traces that corresponded to the different periods of history, but also to changes in land cover. The past aerial photographs proved to be particularly useful in the collection of information from the elderly residents of the area. The aerial photographs of the 1945s and 1970s significantly contributed to the identification of landmarks and specific land covers that no longer exist today. Finally, in the wider study area, information was mapped concerning (a) an ancient path and buildings, (b) the existence of Hellenistic pottery (according to the locals a goddess statue was found in the past; when visiting the fields only a small quantity of Hellenistic pottery was visible on the surface), (c) the location of an Ottoman Krini (in its construction ancient architectural block were used), (d) the existence of a dam in antiquity, and (e) the location of a settlement and a cemetery of the Ottoman period (Figure 5).

The completion of this phase of information collection was followed by temporal, spatial, and thematic categorisation. Then, the information was processed, spatially analysed, and mapped to determine the high-density zones of reference points as well as individual points of discovery of historical elements. A summary of this information collection methodology is shown in the following flow chart (Figure 6).

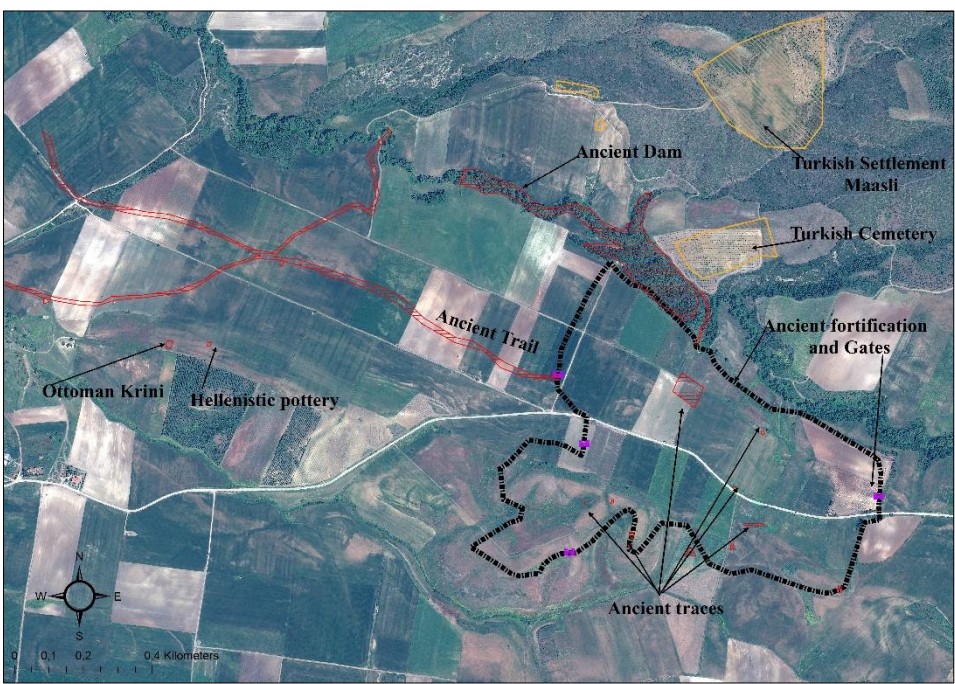

**Figure 5.** Recording of information collected from residents.

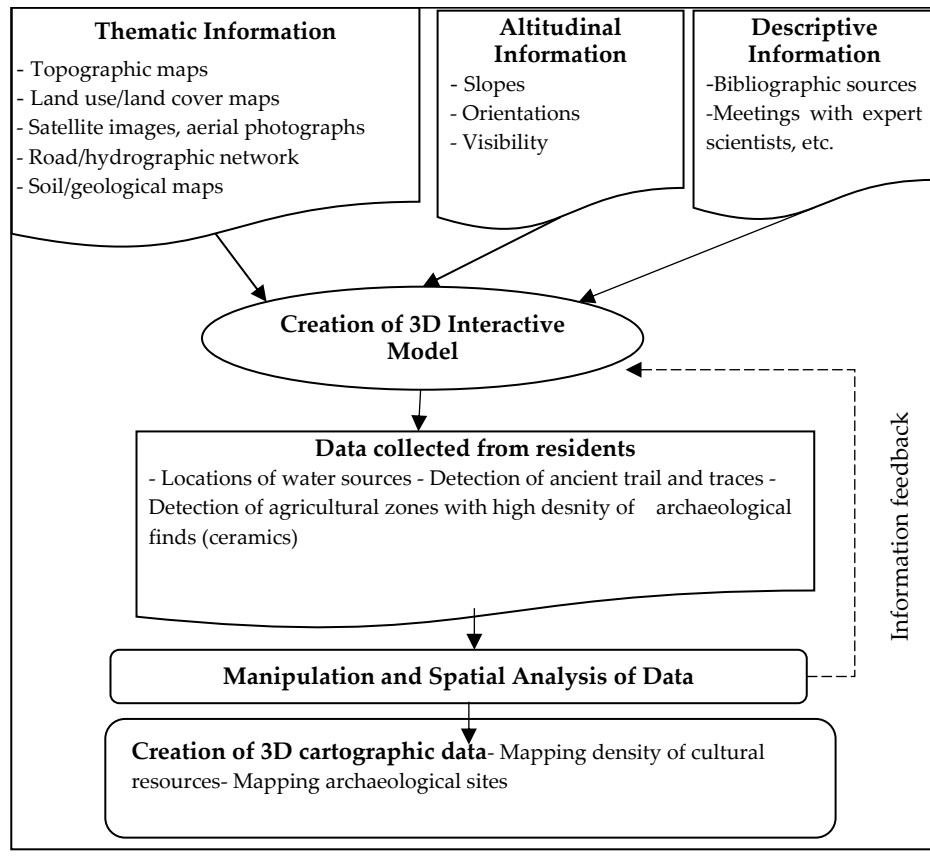

**Figure 6.** Flowchart of 3D GIS model and data analysis.

At the end of the second phase, the findings were evaluated based on the information collected in collaboration with experts such as historians, archaeologists, geographers, etc. The main evaluation indicator was the density of cultural resources taking into account spatial distribution, thematic classification of findings, and their importance. These finding densities are distributed over specific locations within the territorial area. The next step of the investigation using geoinformatics methods was organised within each territorial area in collaboration with the competent authorities.

In this context, the research team, alongside the meetings with residents of the area, applied non-destructive methods and techniques to mark and identify land cover/land use and traces within the archaeological site. Non-destructive methods involve the use of advanced techniques such as (a) processing of satellite images and images from unmanned aerial vehicles (UAVs) at appropriate periods and (b) scanning the subsoil with geophysical surveys.

Multispectral Images and Aerial Photographs as a Means of Identifying Cultural Resources

A multi-spectral satellite image of high spatial resolution on an appropriate date was a key parameter in the current phase of the methodology. Information was collected, either from official sources or with the help of residents, concerning: (a) climatic and meteorological data and (b) the phenological stages of annual crops and in particular, durum wheat. The period between April 20th and May 10th proved to be the most suitable for the purchase of a multispectral image because wheat is in a development phase, thus providing indirect "traces" of any archaeological findings buried at a shallow depth. After a search, an image dated May 4th from the GeoEye-1 satellite, with a spatial resolution of 2 m, was finally acquired. This was followed by the processing of the multispectral image by applying spatial filtering techniques for edge enhancement as well as the creation of vegetation indices (NDVI, normalized difference vegetation index). Archaeological traces were identified in various locations as shown in the image below (Figure 7). The first image on the left shows a ploughed field in October with no archaeological traces. In the same field, with the cultivation of cereals, archaeological traces buried at a relatively shallow depth were detected in May (right image). The identified information was digitised and codified in a geographic information system (GIS).

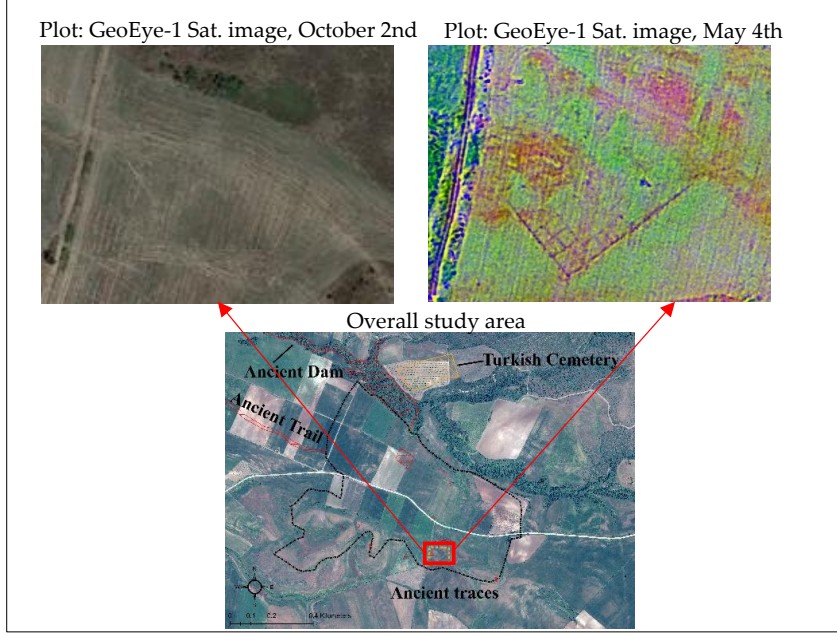

**Figure 7.** Detection and recording of archaeological traces from multispectral satellite images.

Selected zones of the archaeological site were mapped with a UAV from a height of 50 m. The image below depicts part of the outer wall of an ancient city. With aerial photography from a UAV, an orthomosaic map of the archaeological site was created with a spatial resolution of 5 cm. The resulting product was used in collaboration with archaeologists to identify a detailed outline of the wall and its buried or damaged parts (Figure 8).

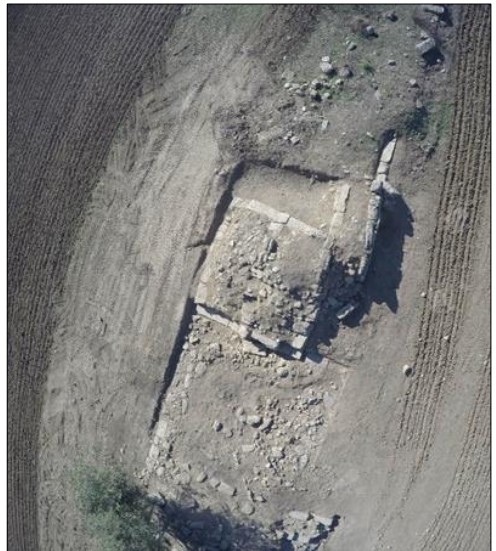

**Figure 8.** Ancient city wall section, mapping from UAV.

The final stage of the methodology consisted of mapping the subsoil using ground-penetrating radar (GPR). This was carried out at the site where the archaeological traces were identified in the analysis of the multispectral satellite image as presented above. The image below shows the field work undertaken as well as the exported result (Figure 9).

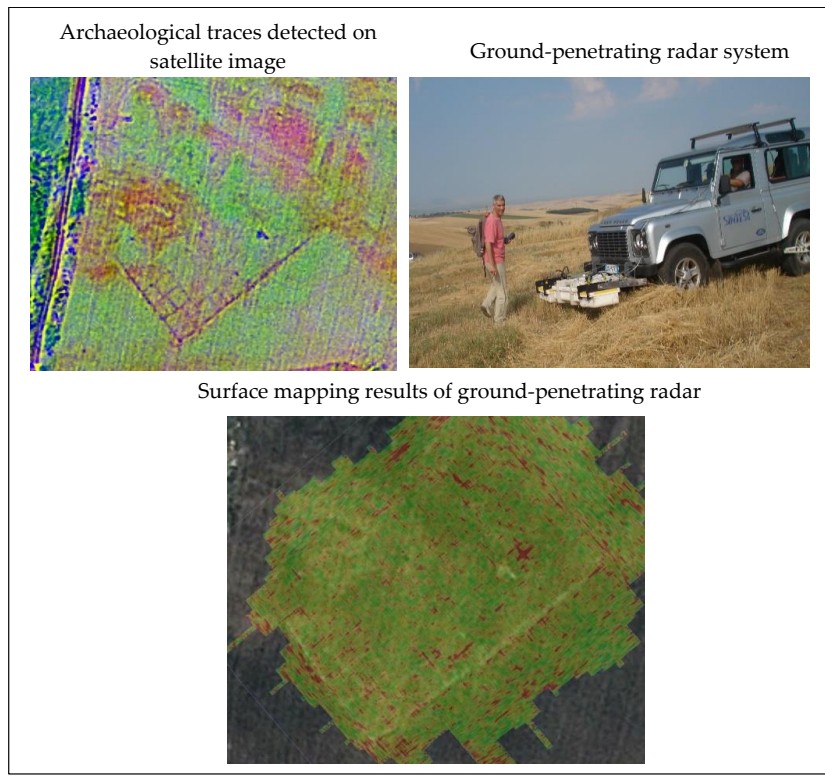

**Figure 9.** Mapping the subsoil using ground-penetrating radar.

Although high-definition ground-penetrating radar was used, the final results do not offer a clear picture of the buried archaeological elements. The archaeological elements appear scattered (shades of red) without forming clear geometric shapes as can be seen in the satellite image. One explanation is increased subsoil humidity resulting in the degradation of the ground-penetrating radar signal. This highlights the importance of using complementary techniques and methods in identifying cultural resources, thus overcoming the inherent weaknesses of these techniques.

## 4. Discussion and Conclusions

### 4.1. Discussion

The proposed methodology is integrated into the process of identifying and valorising territorial resources. Its functionality is based on the fact that it ensures the synergy of three key components: interdisciplinarity, participation, and the use of non-destructive cutting-edge technologies. In practice, this synergy is attained through the functional integration and configuration of methods and tools: (a) social research, with the active participation of the local community as bearers of information, and (b) mapping of the earth's surface and subsurface using interdisciplinary approaches (remote sensing, geophysical surveys, etc.).

The example of identifying and mapping the material elements of cultural resources emphasises the important role of articulation and compatibility of the different technologies and methods used. The key to this is the 3D representations which enhance communication between actors resulting in the activation of collective memory (reconnection between memory and space) [47]. This ultimately leads to the co-production of valuable and implicit information by stakeholders (residents, expert scientists, local bodies, etc.) with the contribution of cutting-edge technologies and the ability to switch between spatial and temporal scales. High-density zones of relevant information emerge in which the high resolution satellite images can focused. The identification of resources within the zones is mapped with ultra-high resolution aerial photographs taken by UAVs. Finally, this is followed by topographic representation, depth, and layout of the traces using ground-penetrating radar.

The following can be considered the main advantages of applying the methodology to this specific field:

- Functional flexibility, with the ability to switch between geospatial analysis scale, as well as the visualisation of all the thematic information over time for a substantiated evaluation, allow the transition:
  - from an individual testimony about the location of a cultural object to collective information for a more precise location in relation to other landmarks,
  - from the scale of the parcel (the point to be identified) to broader territorial units (landscape, geological formations, etc.) for the correlation and the final integration of multiple information sources, and
  - to different time periods, with the help of landscape visualisation, to understand and identify the various land uses/land covers and related landmarks.

- Enhancing transparency, trust and cooperation between the (public) bodies involved and the local community, enabling co-production of information.

Moreover, the proposed methodology is in line with relatively new approaches including digital environmental humanities and biogeocultural–geocultural heritage. The above-referred approaches investigate the way visualization tools and digital data (geodata) could comply with issues related to humans and the environment [48]. Especially in the framework of biocultural heritage, as defined by UNSECO, the proposed methodology could be applied to local rural communities and indigenous people, aiming to retrieve vital information, concerning knowledge, and practices being followed by them in order to succeed in sustainable use of their local resources [49,50]. Finally, the geo-tools applied by the integrated methodology (3D GIS models, remote sensing, and Drone mapping) could comprise the main components in the virtual heritage development. In a "Virtual Heritage"

approach, 3D representations may assist in the comprehension of the complex nature of geoheritage and its multiple connections [51].

*4.2. Conclusions*

The application of this combined methodological approach provides a comprehensive view of the total cultural resources in an area utilising information accumulated from multiple sources and scales. At the same time, the application of remote sensing techniques and geophysical surveys helps verify and document the information collected, either from bibliographic sources or the oral testimonies of residents. More specifically, the methodology proved to be innovative in terms of identifying archaeological traces as it reverses the predominant investigation process, which usually starts from a random excavation point (intersection) with high chance of failure. It also offers the competent authorities and local government, a layout plan of the traces (with spatial reference and distribution), contributing to the evaluation, and selection of the starting points for excavation projects. This accelerates the process of excavating and identifying archaeological resources.

In conclusion, the application of the proposed methodology contributes (a) to the creative combination of information drawn from the local community, accumulated over time and implicitly, with the help of tools based on new technologies (GIS-remote sensing), (b) to identifying the overall cultural resources and their links to the local community, (c) to reducing the time and cost of the research process, and (d) to ensuring a strong consensus between endogenous and exogenous stakeholders, which is considered crucial for the implementation of a territorial diagnosis.

Furthermore, the above analysis shows that applying this methodology leads to the integrated valorisation of cultural and archaeological resources within a broader local territorial development plan. Such a plan could now (a) enhance the contribution of cultural heritage to specificity and increase the added value of territorial resources, and (b) contribute to the reconstruction of local histories in terms of the relationship between people and places (settlement, landscapes, etc.). In essence, the methodology addresses the increasingly specialised needs of territorial plans for the valorisation of cultural resources. Finally, it supports the active participation of residents through the combined and effective use of new technological tools, while fostering the role of cultural heritage in building the identity and image of small-scale territorial areas.

Despite the benefits occurring from the combined methodology, some prerequisites have emerged. The scientific team should be multidisciplinary and aware of a number of spatial components concerning the study area, i.e., land cover, cultivation growing stage, and climatic and meteorological conditions. The team should possess deep knowledge of satellite image processing and ground-penetrating survey methodologies, keeping in mind that GPR equipment is very expensive.

Future studies should be focused on the 3D GIS models' flexibility by their incorporation in open-source GIS software and the availability of high-resolution geospatial data. This will make feasible the proposed methodology to apply in small local communities that face spatial problems.

**Author Contributions:** D.G. and I.F. conceptualized the research, performed the validation, and wrote the manuscript. All authors have read and agreed to the published version of the manuscript.

**Funding:** This research received no external funding.

**Acknowledgments:** The triggering for this study was the collaboration among the University of Thessally (Greece), Antiquities Ephorate of Larissa (Greece), and University of Messina (Italy) in the framework of the Greek–Italian Archaeological program in the Skotoussa of Farsala.

**Conflicts of Interest:** The authors declare no conflict of interest.

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
