# Peer review of "Integrated Remote Sensing and 3D GIS Methodology to Strengthen Public Participation and Identify Cultural Resources"

_land, doi:10.3390/land11101657_

Round 1

Reviewer 1 Report

Formally, some details may need reviewing. A few examples: FARO is not an acronym but a city in Portugal, so no need to capitalize each letter. Also, although it is not really a syntactic issue, there is a number of word repetitions in close proximity (the same 1-2 lines), something that can easily be solved, making the text more pleasant to read – again, this is a matter of style, not content, so I point this out mainly as a suggestion. But a thorough review is needed. For instance, “[11] notes that…” reflects a transition to the MDPI citation system without maintaining coherence (e.g. “Kaeser [11] notes that…”).

In terms of substance, the subject of activating collective memory, partially through technological applications, comes over as interesting. Two of the methodological steps are straightforward, namely the use of existing sources and data, on the one hand, and the remote sensing and geophysical surveys, on the other. These are commonplace in any territorial study, not only in heritage and archaeology but in the wider landscape studies as well (geography, engineering, economics, …). The differentiating element is the community integration into what usually is a merely technical exercise. One potential weakness is the selection of the interlocutors, as they seem to be selected based on their reputation of having proper knowledge on the before-and-afters of local heritage. One may question if they are indeed representative of any community – a fundamental caveat in oral history.

I do sense there is great potential in exploring this sort of convergence between landscape archaeology, local history, and community involvement. I therefore commend the authors for their paper, which however falls a bit short in demonstrating to an external reader what the added value of involving local people really was, in clear terms. The conclusion states that this research demonstrated the integrated valorization of archaeology for territorial development. Most will agree with this premise, but the text would benefit quite a lot from an additional paragraph not filled with theory or generic statements, but on the effective outcomes of involving local residents. Are they useful merely to indicate/confirm archaeological sites on old photos and 3D renderings? Or are they part of a complex, systemic interaction between public and private cultural stakeholders? When I started reading the paper I was expecting much more attention given to this sociological angle, and the case study might have explored this more in detail. But this is still a nice reflection that mentions the community layer in landscape archaeology.

Author Response

Dear reviewer thanks for the intuitive comments.

1)  The mistakes such us, FARO and Kaeser [11] ....have been corrected. Word repetitions namely "territorial" have been minimized where it was possible.

2) About representative of any community: The main criterion of choice  of interlocutors was based on the insights about the sites and the positions of cultural heritages (ancient traces or findings). 

3) About the effective outcomes of involving local residents:  The participants offer their knowledge about the ancient traces. Some reflections of the participants have been added in "results" section, lines 394-400

Reviewer 2 Report

This article is a well-edited and interesting study. 

The importance of the topic is shown by the many references used.

A good example of the use of remote sensing methods is presented. 

Of particular note is the emphasis on combining methods, indicating that sociological and historical methods and modern GIS applications can be produce new results if combined them.  

The choice of sample area is also interesting, albeit small. 

The structure of the study is good, but the proportions are a little exaggerated in the direction of theory. 

It is recommended to shorten the Introduction and Methods chapters and to omit the more general information.

The Results chapter illustrates well the potential of combining local community information with remote sensing methods. 

The findings are well substantiated. 

Of particular note is the attention drawn to the use of 3D visualisation tools, which can make the investigations more interesting for the (local) population, thus mobilising them to search for information.

The lists should be standardised in the editing (some are in the text (lines 76, 78) , some are highlighted paragraphs).

Author Response

Dear reviewer thanks for your comments and suggestions 

1) About to shorten the Introduction and Methods chapters: In "Introduction" section, lines 63 & 65 (same words) and lines 143-146 have been deleted. 

2) Standardised lists: The lists have been standardised throughout the article. 

Reviewer 3 Report

This article presents an extremely novel research, which proposes an integrated methodology of remote sensing and 3D GIS for the identification of cultural resources. As stated by the authors, the approach is interdisciplinary, participatory with the local community and promotes the use of non-destructive state-of-the-art technologies.

In order to demonstrate the applicability of the methodology, a case study in the ancient city of Skotoussa in Greece is addressed. 

One of the most significant contributions of this research is methodological. Each step and decision is exhaustively explained. However, it is recommended to the authors to incorporate a figure that allows understanding the methodological design in a simple and direct way. 

In the results section, it is suggested that the researchers could incorporate some of the reflections of the participants of the heritage workshops. A qualitative description of their perceptions and narratives would provide a better understanding of how people activate their knowledge of heritage.

The article has a number of original and very relevant ideas, however, it does not refer to the global literature that has addressed these issues, or developed similar research. I believe that this article could expand its impact much more, if it incorporates some recent theoretical and methodological aspects that are incorporating interdisciplinarity and technologies for territorial research. Particularly the digital environmental humanities approach, which promotes the use of emerging technologies to address contemporary challenges encountered by inhabitants in the face of widespread environmental change. In this field of study, the concept of virtual heritage emerges.

Similarly, it is suggested that the authors at least discuss the concept of biogeocultural heritage and geocultural heritage, which integrates the geographic dimension of heritage. 

In the conclusions, it is recommended that the authors detail the problems and difficulties encountered in applying the method. Likewise, to deepen in the recommendations that can be given to future studies. 

In line no. 49, you should be more specific about what you mean by: "in line with the trend of transition from public goods to common goods".

Figures 2, 3, 4, 6 and 8 should be improved. They have poor resolution. A design improvement of the figures can go a long way in making the results of the article known. 

Author Response

Dear reviewer thank you for the  comments and suggestions.

1) Incorporate a figure at Methodological sector: The figure has been added (lines 200-201).

2) Reflections of the participants: At the "result" section some  reflections of the participants have been added (lines 394-399).

3) Incorporating methodological aspects such as digital environmental humanities approach and biogeocultural heritage and geocultural heritage: At the "discussion" section a paragraph has been added about all the above mentioned approaches and how the proposed methodology could contribute to them.  Also relevant references have been added (lines 514-525). 

4) Problems, recommendations: At the "conclusions" two paragraphs about difficulties encountered in applying methodology and future studies have been added (lines 557-566). 

5) line no 49, "in line with the trend of transition from public goods to common goods": Because the state tends no longer to bear the cost of public goods, the territory intervenes more and more often to regulate, manage and turn into common goods its own resources of great interest.

6) Figures 2, 3, 4, 6 and 8 should be improved: Most of the figures have been improved.